# Self-Esteem as a Predictor of Mental Adjustment in Patients with Breast Cancer

**DOI:** 10.3390/ijerph182312588

**Published:** 2021-11-29

**Authors:** Pei-Ling Tsai, Ting-Ting Kuo, Chih-Hung Ku, Guo-Shiou Liao, Chi-Kang Lin, Hsueh-Hsing Pan

**Affiliations:** 1Department of Nursing, Tri-Service General Hospital, Taipei City 11490, Taiwan; 609209004@mail.ndmctsgh.edu.tw; 2Emergency Room, Department of Nursing, Cheng Hsin General Hospital, Taipei City 112401, Taiwan; chf106048@chgh.org.tw; 3School of Public Health, National Defense Medical Center, Taipei City 11490, Taiwan; cku@mail.ndmctsgh.edu.tw; 4General Surgery, Division of Surgery, Tri-Service General Hospital, National Defense Medical Center, Taipei City 11490, Taiwan; guoshiou@mail.ndmctsgh.edu.tw; 5Department of Gynecology and Obstetrics, Tri-Service General Hospital, National Defense Medical Center, Taipei City 11490, Taiwan; kung568@mail.ndmctsgh.edu.tw; 6School of Nursing, National Defense Medical Center, Taipei City 11490, Taiwan

**Keywords:** breast cancer, self-esteem, self-respect, self-regard, mental adjustment, psychological adjustment

## Abstract

This study aimed to explore the relationship between self-esteem and mental adjustment and examine the directional effects in patients with breast cancer using path modeling. This was a cross-sectional, descriptive, and correlational study. A total of 128 patients with breast cancer were selected through convenience sampling at a medical center in northern Taiwan. They completed a basic characteristics questionnaire, the Memorial Symptom Assessment Scale short form, the Rosenberg Self-Esteem Scale, and the mini-Mental Adjustment to Cancer Scale. Descriptive statistics, regression analysis, and path analysis were used to analyze the data. The results showed that higher self-esteem was associated with better mental adjustment (β = 0.9, 95% confidence interval 0.6~1.3, *p* < 0.001). Age, religious beliefs, employment, cancer stage, and symptom distress were correlated with mental adjustment. Path modeling demonstrated that self-esteem, cancer stage, performance status, and symptom distress directly affected mental adjustment in patients with breast cancer. These findings suggest that health professionals should evaluate self-esteem, performance status, and symptom distress in patients with breast cancer immediately upon admission. This can facilitate early implementation of relevant nursing interventions and, consequently, improve self-esteem and symptom distress and increase mental adjustment in these patients.

## 1. Introduction

About 2.3 million women were newly diagnosed with breast cancer and 685,000 died globally in 2020 [1]. In Taiwan, breast cancer has the highest incidence rate among all cancers in women. There were 16,608 women with breast cancer, accounting for 88% of the total cancer incidence in 2017. The mortality rate of female breast cancer was the fourth highest among the top 10 cancers, at 22.3 per 100,000 population in 2020 [2]. Breast cancer is a stressful event that causes extremely difficult physical, emotional, and social challenges. In addition to worrying about whether breast cancer will cause metastasis and the impact of the side effects of treatment on daily life, patients with breast cancer are more likely to experience depression than patients with other cancers [3]. In addition, surgery, chemotherapy, radiation therapy, and other forms of treatment can cause changes in the patient’s physical appearance, leading to anxiety, pain, depression, and low self-esteem [4].

Self-esteem is determined by positive or negative feelings as well as satisfaction and confidence in oneself. It also refers to the belief that one “is capable of coping with the challenges in life and is worthy of happiness” [5]. One study showed that self-esteem is a key factor in the growth and return to normal life in patients with breast cancer [6]. Low self-esteem has been found to be strongly correlated with depression [7] and other psychological distress [8]. Mental adjustment to cancer (MAC) is a psychological coping response to cancer that involves the adoption of strategies to deal with life-threatening situations [9]. This adjustment can be maladaptive or adaptive and is affected by the patient’s age, personality traits, religious attitudes, family support, social contexts, and the attitudes of their family and health care providers [10,11]. Studies indicate that maladaptive mental adjustment such as helplessness–hopelessness is negatively correlated with physical, emotional, and functional well-being [12], increases with anxiety and depression [13], and affects quality of life [11,12].

A meta-analysis related to self-esteem focused on the trajectory of self-esteem across the life span [14]. A systematic review examined the factors associated with hope and hopelessness in patients with cancer [15]. There have been no systematic reviews or meta-analyses exploring the relationship between self-esteem and mental adjustment. Previous studies have discussed the relationship between self-esteem and mental adjustment [16,17]. However, these two studies focused on children and not on patients with breast cancer. A prior study selected patients with breast cancer to explore the effects of the psychological factors of self-esteem and optimism on the relationship with quality of life [18]. Nevertheless, this study did not focus on the relationship between self-esteem and mental adjustment. In addition, one previous study explored self-esteem and mental adjustment in patients with breast cancer. However, this study focused on the effects of psychological interventions [19]. Therefore, the relationship between self-esteem and mental adjustment remains unclear, especially in patients with breast cancer. Moreover, previous studies have only applied multiple regression models to identify the relationship between self-esteem and mental adjustment. We included path analysis and regression analysis in our evaluation. Path analysis allows for the observation of direct and indirect effects concurrently with multiple independent and dependent variables. It is also a method for examining directional patterns among a set of variables [20]. Therefore, the present study aimed to explore the relationship between self-esteem and mental adjustment and examined the directional effects in patients with breast cancer using path modeling.

## 2. Methods

### 2.1. Study Design and Participants

This was a cross-sectional study with convenience sampling from the general ward of a medical center in northern Taiwan from October 2019 to March 2020. Through a power analysis with significance set at 0.05, power of 0.8, and effect size of 0.15, using G*Power 3.1.9.2 (https://www.psychologie.hhu.de/arbeitsgruppen/allgemeine-psychologie-und-arbeitspsychologie/gpower, accessed on 20 August 2021) [21], it was determined that a sample size of 101 was needed. We originally approached 138 hospitalized patients with breast cancer for inclusion in the study. The inclusion criteria were aged 18 years or more, free of cognitive impairments or mental illness, able to communicate in Mandarin or Taiwanese, and agreed to participate in the study. Ten patients were either too weak to continue the study or dropped out for other reasons, and thus, were excluded from the final study. Therefore, a total of 128 participants completed the study.

### 2.2. Instruments

#### 2.2.1. Demographics and Disease- and Treatment-Related Characteristics

Demographic characteristics included participant age, educational level (junior high school or below, high school, or specialist or above), having religious beliefs (yes or no), marital status (single or married), parity, child (yes or no), living with others (yes or no), caregiver (yes or no), economic status (<19,999 NT dollars, 20,000~49,999 NT dollars, or 50,000 NT dollars or above), and employment status (unemployed, unemployed due to disease, employment change due to disease, or employed). Disease-related characteristics included duration of cancer diagnosis, cancer stage, comorbidity (yes or no), and Eastern Cooperative Oncology Group (ECOG) performance status (0 = fully active; 1 = restricted in physically strenuous activity but ambulatory; 2 = ambulatory and capable of all self-care; 3 = only limited self-care; 4 = completely disabled; and 5 = dead) [22]. Data were also collected on treatment-related characteristics, with or without breast cancer-related surgery (e.g., chemotherapy, radiotherapy, target therapy, hormone therapy, and immune therapy).

#### 2.2.2. Memorial Symptom Assessment Scale Short Form (MSAS-SF)

The Memorial Symptom Assessment Scale was developed by Portenoy et al. [23], and a shortened version (the MSAS-SF) was developed by Chang et al. [24]. A Chinese version of the scale was developed and reported to have good validity and reliability by Lam et al. [25]. It contains 32 items, 28 on physical symptoms, and four on psychological symptoms, which are used to evaluate patient self-reported symptom distress during the past week. Each symptom is rated on a 5-point scale (0 = “no distress” and 4 = “very much”. The total score ranges from 0 to 128, with a higher score indicating more distress [25]. In the present study, Cronbach’s α was 0.945 for 128 patients with breast cancer.

#### 2.2.3. Rosenberg Self-Esteem Scale (RSES)

The Rosenberg Self-Esteem Scale (RSES) was used to measure the patients’ self-reported self-esteem level. It contains 10 items. Each item is rated on a 4-point scale, ranging from 1 to 4 (1 = “very disagree” and 4 = “very agree”. The reverse items of 2, 5, 6, 8, and 9 were transformed into the positive items. The total score ranges from 10 to 40, with higher scores indicating higher self-esteem. The Cronbach’s α was 0.87 in a large sample of junior high and high school students [26]. In this study, the Cronbach’s α for self-esteem was 0.870 in 128 patients with breast cancer.

#### 2.2.4. Mini-Mental Adjustment to Cancer Scale (Mini-MAC)

The Mini-Mental Adjustment to Cancer Scale (Mini-MAC) was developed by Watson et al. [27] and has many different versions [9,28,29,30,31]. In this study, we used the Chinese version of the scale [32]. Mini-MAC measures the self-reported coping response of patients with cancer. It contains 29 items divided into five subscales: helpless and hopeless (HH, 8-item), anxious preoccupation (AP, 8-item), fighting spirit (FS, 4-item), cognitive avoidance (CA, 4-item), and fatalism (FA, 5-item). Each item is rated on a 4-point scale (0 = “definitely does not apply to me”, and 3 = “definitely applies to me”. The HH, AP, and CA subscales concern the more impassive coping strategies, and FS and FA concern the more active coping strategies. The subscales on impassive coping strategies are converted and combined with active coping strategies to calculate a total score for mental adjustment, which ranges from 0 to 87 with higher scores representing higher MAC. The Mini-MAC has been shown to have good reliability and validity in patients with cancer in prior study [12]. The Cronbach’s α of the Mini-MAC was 0.814 for 128 patients with breast cancer in this study.

### 2.3. Study Procedure

This study was performed after approval from the institutional review board of the authors’ institution (approval no. 2-108-05-029). Patients with breast cancer who met the inclusion criteria were asked to participate and were given information on the study protocol. The researcher explained the objectives and methods of the study to the patients in a meeting room. After written informed consent was obtained, the researcher collected the data face to face via questionnaires. The questionnaires were anonymous, and the collected information was confidential. Patients completed the questionnaires in 15–20 min and were informed that they were allowed to withdraw from the study at any time.

### 2.4. Statistical Analysis

Data were analyzed using IBM SPSS Statistics for Windows, Version 22.0 (IBM Corp., Armonk, NY, USA). Frequency and percentage were used to describe categorical variables, and mean and standard deviation (SD) were used to express continuous variables. Pearson correlation was used to analyze the relationship between self-esteem and mental adjustment along with its dimensions. Multiple linear regression was used to define the predictors of mental adjustment. We used path analysis to describe the direct or indirect dependencies among a set of variables including patient characteristics, disease-related characteristics, and treatment-related characteristics. A *p*-value of <0.05 was considered statistically significant.

## 3. Results

The mean age of the participants was 53.6 years. Most of the patients had a specialist or above level of education (47.7%). The majority of the participants had religious beliefs (75.0%), were married (69.5%), had two parities (34.4%), had children (76.6%), lived with others (89.8%), had caregivers (89.1%), had an income lower than 19,999 NT dollars per month (40.6%), were unemployed due to disease (35.9%), had stage II cancer (38.3%), had no comorbidity (72.7%), had an ECOG status of 0 (59.4%), and received cancer-related surgery with or without other treatments (75.8%). The mean years since cancer diagnosis was 2.5, and the mean symptom distress level was 21.2 (Table 1).

The mean score for self-esteem was 29.8 (SD = 5.1). The mean score for mental adjustment was 56.1 (SD = 10.1). The mean scores for the HH, AP, FS, CA, and FA subscales of the Mini-MAC were 5.9 (SD = 4.7), 10.5 (SD = 5.4), 8.8 (SD = 2.1), 7.1 (SD = 2.7), and 8.6 (SD = 2.2), respectively. Self-esteem was significantly and positively correlated with mental adjustment (*r* = 0.584, *p* < 0.001). As self-esteem increased, the mental adjustment level improved in the patients. We transformed the negative subscales (HH, AP, and CA) into positive items. The results showed that self-esteem was significantly correlated with HH (*r* = −0.730, *p* < 0.001), AP (*r* = −0.489, *p* < 0.001), FS (*r* = 0.358, *p* < 0.001), and FA (*r* = −0.205, *p* = 0.020) dimensions. The higher the level of self-esteem, the lower the level of HH, AP, and FA, and the higher the level of FS in patients with breast cancer (Table 2).

As shown in Table 3, self-esteem, age, having religious beliefs, employment status, cancer stage, and level of symptom distress were significant predictors of mental adjustment in the patients after adjustment for their demographic and disease-related characteristics. Patients with a higher mean score for self-esteem (β = 0.9, 95% confidence interval [CI] = 0.6~1.3, *p* < 0.001), older age (β = 0.3, 95% CI = 0.1~0.5, *p* = 0.04), and lower symptom distress (β = −0.2, 95% CI = −0.3~−0.1, *p* < 0.001) had more effective mental adjustment. Patients who had religious beliefs had a higher mean score for mental adjustment than those who had no religious beliefs (β = 3.7, 95% CI = 0.6~6.9, *p* = 0.024). Patients who were employed had a higher mean score for mental adjustment than those who were unemployed (β = 4.1, 95% CI = 0.2~7.9, *p* = 0.040). Patients who had been diagnosed as stage III had a lower mean score for mental adjustment than those who had been diagnosed as stage I (β = −8.0, 95% CI = −12.7~−3.3, *p* < 0.001).

As shown in Table 4, path modeling demonstrated that self-esteem, cancer stage, performance status, and symptom distress directly affected mental adjustment. The relationships between self-esteem (coefficients = 0.584, *p* < 0.001), diagnosis as stage III compared with I (coefficients = −0.232, *p* = 0.010), performance status, ECOG status of I compared with 0 (coefficients = −0.215, *p* = 0.017), and symptom distress (coefficients = −0.546, *p* < 0.001) were significant by standardized coefficient estimates for the paths. However, these factors did not affect mental adjustment indirectly through self-esteem.

## 4. Discussion

To our knowledge, this study is one of the few to explore the relationship between self-esteem and mental adjustment in patients with breast cancer through regression and path modeling. We found that self-esteem was significantly and positively correlated with mental adjustment in patients with breast cancer. Self-esteem is an important personal coping ability that can mitigate pressure from life events in cancer patients. Self-esteem is also a dynamic process of self-evaluation and is closely related to mental adjustment [33]. Cancer patients with low self-esteem are likely to have worse mental adjustment; they may be worried about the adverse effects of their diagnosis or treatment and feel unprotected, leading to the inability to deal with mental adjustment [34].

The results showed that older patients had better mental adjustment. This finding is consistent with those of previous studies [35,36]. This may be because younger patients with breast cancer are depressed about being diagnosed with cancer and need to overcome the fear of treatment [37]. As they become older, patients have more life experiences and think they are ready to face death. However, one prior study found that the older the patient, the worse the level of mental adjustment [36]. It may be that, the older the patient with breast cancer, the greater the risk of comorbidities caused by chronic diseases or treatment compared with young patients with breast cancer, leading to the finding that the higher the age, the worse the degree of psychological adjustment [37]. Patients with religious beliefs had better mental adjustment. One study indicated that religious beliefs were positively correlated with fatalism, and negatively correlated with helplessness, hopelessness, and anxiety [38]. Religious beliefs have a profound impact on daily life and are related to all aspects of personal experience [38]. Cognitive mechanisms were used to influence mental adjustment and considered sources of support that can help individuals overcome difficulties in life [39]. Therefore, religious beliefs can enable them to actively cope by reducing depression, increasing their adjustment capabilities, and enhancing their own and their family’s well-being [38]. This study showed that patients who were employed had higher mental adjustment than those who were unemployed. Patients who were employed were able to maintain their income, independence, and bond with society. It was also indicated that these patients had good performance status, and physical functional ability to perform their work. One prior study showed that patients with breast cancer who returned to work had better body image and sexual function, less disability and pain, and lower anxiety and depression [40].

The findings of this study indicated that patients with stage I breast cancer had better mental adjustment than those with stage III breast cancer. This was similar to the results of a previous study, wherein patients with stage IV breast cancer had higher levels of anxiety than those with stage I breast cancer [39]. This finding may be because of the higher symptom burden and poor physical function in advanced patients [39]. Patients with advanced breast cancer may experience recurrence or metastasis, and anxiety is more common in this population. It may be that, because of the severity of the disease and treatment, the patient is more likely to hide the prognosis from family members and friends, and, as a result, experience isolation and loss of control, which increases the level of anxiety [41]. In addition, patients experienced side effects during the long treatment process and required a considerable amount of energy to cope with it. This may lead to a decline in the degree of mental adjustment in patients with breast cancer [42]. Patients with breast cancer with lower symptom distress exhibited better mental adjustment. This finding is similar to that of a previous study [43]. The degree of mental adjustment in patients with breast cancer has been shown to affect their levels of anxiety, depression, and quality of life [44]. The symptoms and emotional distress of patients with breast cancer are not static. The greater the physical and psychological symptom distress, the poorer the coping strategies in patients with breast cancer [45]. From the results of path modeling, this study indicated that self-esteem, cancer stage, performance status, and level of symptom distress can directly affect mental adjustment. One prior study showed that performance status is highly correlated with mental adjustment to cancer [46]. Performance status indicated patients’ physical functional status as well as the autonomy of their body. Therefore, deteriorated functional status is strongly associated with poor psychological outcomes in patients with cancer.

This study had several limitations. First, this was a cross-sectional study, and a longitudinal study is needed to prove the causal relationship between self-esteem and mental adjustment. Second, the participants in this study were all hospitalized patients, and this study did not include outpatients with breast cancer, which may have resulted in an overestimation of disease severity. Third, this study used convenience sampling, which may limit the generalizability of the findings. There are several avenues for future research. First, future studies should explore the relationship between self-esteem and mental adjustment in patients with breast cancer using longitudinal designs. Second, future studies should expand the study population to include outpatients with breast cancer to compare the differences in self-esteem and mental adjustment between inpatients and outpatients. Third, random sampling should be performed to prevent selection bias and increase the generalizability of the findings.

## 5. Conclusions

This study found that self-esteem was positively correlated with mental adjustment in patients with breast cancer. Older people who had religious beliefs, were employed, had a lower stage of cancer, and had lower symptom distress showed more effective mental adjustment. Patients’ self-esteem, cancer stage, performance status, and symptom distress were found to directly affect mental adjustment. We recommend that future interventions focus on evaluating self-esteem and performance status as well as lowering symptom distress in these patients immediately upon admission. Thus, health professionals can implement relevant nursing interventions to improve patients’ self-esteem and symptom distress and increase mental adjustment.

## Figures and Tables

**Table 1 ijerph-18-12588-t001:** Characteristics of demographics, disease-related, and treatment among patients with breast cancer (*n* = 128).

Variables	Range	Mean	±SD/*n* (%)
Demographics			
Age	26.5–79.3	53.6	±9.4
Educational level			
Junior high school below		27	(21.1)
High school		40	(31.3)
Specialist or above		61	(47.7)
Religious belief			
No		32	(25.0)
Yes		96	(75.0)
Marital status			
Single		39	(30.5)
Married		89	(69.5)
Parity			
0		27	(21.1)
1		22	(17.2)
2		44	(34.4)
≥3		35	(27.3)
Child			
No		30	(23.4)
Yes		98	(76.6)
Living with others			
No		13	(10.2)
Yes		115	(89.8)
Caregiver			
No		14	(10.9)
Yes		114	(89.1)
Economic status			
<19,999 NT dollars		52	(40.6)
20,000~49,999 NT dollars		51	(39.8)
50,000 NT dollars		25	(19.5)
Employment			
Unemployed		33	(25.8)
Unemployed due to disease		46	(35.9)
Employed change due to disease		13	(10.2)
Employed		36	(28.1)
Disease-related characteristics			
Duration of cancer diagnosis (year)	0–20.8	2.5	±3.6
Cancer stage			
I		23	(18.0)
II		49	(38.3)
III		16	(12.5)
IV		40	(31.3)
Comorbidity			
No		93	(72.7)
Yes		35	(27.3)
Performance status			
ECOG = 0		76	(59.4)
ECOG = 1		39	(30.5)
ECOG ≥ 2		13	(10.2)
Characteristics of treatment			
Treatment following cancer diagnosis			
Cancer operation ± other treatments		97	(75.8)
Other treatments		31	(24.2)
Symptom distress		21.2	±22.4
Psychological symptom		3.8	±4.1
Physical symptom		17.4	±19.0

**Table 2 ijerph-18-12588-t002:** Correlation of self-esteem, mental adjustment, and its dimensions among patients with breast cancer (*n* = 128).

Variables	Mean	SD	*r*	*p*-Value
Self esteem	29.8	5.1		
Mental adjustment	56.1	10.1	0.584	<0.001
Helpless and Hopeless	5.9	4.7	−0.730	<0.001
Anxious Preoccupation	10.5	5.4	−0.489	<0.001
Fighting Spirit	8.8	2.1	0.358	<0.001
Cognitive Avoidance	7.1	2.7	−0.165	0.063
Fatalism	8.6	2.2	−0.205	0.020

**Table 3 ijerph-18-12588-t003:** Predictors of mental adjustment among patients with breast cancer (*n* = 128).

Variables	Univariate Analyses		Multivariate Analysis	
	Crude β (95% CI)	*p*-Value	Adjusted β (95% CI)	*p*-Value
Self-esteem	1.2 (0.9~1.5)	<0.001	0.9 (0.6~1.3)	<0.001
Demographics				
Age	0.1 (−0.1~0.3)	0.415	0.3 (0.1~0.5)	0.014
Educational level				
Junior high school below	Reference		Reference	
High school	−0.3 (−3.8~3.3)	0.878	1.6 (−2.6~5.8)	0.455
Specialist or above	1.0 (−2.8~4.8)	0.593	0.42 (−3.4~4.3)	0.832
Religious belief				
No	Reference		Reference	
Yes	3.6 (−0.4~7.6)	0.083	3.7 (0.6~6.9)	0.024
Marital status				
Single	Reference		Reference	
Married	0.3 (−3.5~4.1)	0.873	0.2 (−3.2~3.6)	0.918
Parity				
0	Reference		Reference	
1	2.4 (−1.5~6.32	0.235	1.9 (−8.1~11.9)	0.714
2	0.3 (−3.4~ 4.0)	0.880	1.51 (−8.7~11.7)	0.772
≥3	−1.0 (−5.7~3.7)	0.672	0.7 (−8.8~10.1)	0.894
Child				
No	Reference		Reference	
Yes	2.6 (−1.5~6.7)	0.218	1.3 (−8.0~10.5)	0.790
Living with others				
No	Reference		Reference	
Yes	0.2 (−5.6~6.1)	0.936	0.5 (−4.7~5.7)	0.842
Caregiver				
No	Reference		Reference	
Yes	−1.1 (−6.8~4.5)	0.694	−1.0 (−5.5~3.5)	0.654
Economic status				
<19,999 NT dollars	Reference		Reference	
20,000~49,999 NT dollars	1.5 (−3.0~5.9)	0.519	−1.1 (−5.9~3.7)	0.659
50,000 NT dollars	0.5 (−3.1~4.1)	0.779	−2.5 (−5.7~0.6)	0.116
Employment				
Unemployed	Reference		Reference	
Unemployed due to disease	2.5 (−1.4~6.4)	0.210	2.7 (−1.7~7.1)	0.235
Employed change due to disease	−1.1 (−6.9~4.7)	0.712	2.8 (−2.6~8.2)	0.308
Employed	−0.1 (−3.8~3.6)	0.958	4.1 (0.2~7.9)	0.040
Disease-related characteristics				
Duration of cancer diagnosis (year)	0.4 (−0.1~0.9)	0.084	0.3 (−0.1~0.7)	0.191
Cancer stage				
I	Reference		Reference	
II	1.9 (−1.7~5.5)	0.308	−0.8 (−4.5~2.9)	0.669
III	−7.1 (−12.3~−1.9)	0.008	−8.0 (−12.7~−3.3)	0.001
IV	1.0 (−2.8~4.8)	0.606	1.8 (−2.3~5.8)	0.388
Comorbidity				
No	Reference		Reference	
Yes	−0.01 (−4.0~3.9)	0.997	−3.0 (−6.1~0.1)	0.061
Performance status				
ECOG = 0	Reference		Reference	
ECOG = 1	−7.2 (−12.9~−1.5)	0.015	0.3 (−5.0~5.6)	0.914
ECOG ≥ 2	−0.9 (−4.8~2.9)	0.631	1.0 (−1.9~3.9)	0.505
Characteristics of treatment				
Treatment following cancer diagnosis				
Cancer operation ± other treatments	−0.9 (−5.0~3.2)	0.671	−1.4 (−4.7~1.9)	0.417
Other treatments	Reference		Reference	
Symptom distress	−0.3 (−0.3~−0.2)	<0.001	−0.2 (−0.3~−0.1)	<0.001

**Table 4 ijerph-18-12588-t004:** Path modeling of self-esteem and mental adjustment in breast cancer patients (*n* = 128).

Variables	Self-Esteem → Mental Adjustment		Mental Adjustment	
	Coefficients	*p*-Value	Coefficients	*p*-Value
Self esteem			0.584	<0.001
Age	0.142	0.114	0.073	0.414
Educational level				
Junior high school below	Reference		Reference	
High school	−0.102	0.254	−0.014	0.875
Specialist or above	0.071	0.426	0.048	0.591
Religious belief				
No	Reference		Reference	
Yes	0.135	0.132	0.154	0.086
Marital status				
Single	Reference		Reference	
Married	0.019	0.834	0.014	0.875
Parity				
0	Reference		Reference	
1	0.054	0.549	0.106	0.236
2	0.048	0.591	0.013	0.884
≥3	−0.022	0.804	−0.038	0.670
Child				
No	Reference		Reference	
Yes	0.109	0.224	0.110	0.219
Living with others				
No	Reference		Reference	
Yes	−0.028	0.757	0.007	0.937
Caregiver				
No	Reference		Reference	
Yes	−0.032	0.718	−0.035	0.695
Economic status				
<19,999 NT dollars	Reference		Reference	
20,000~49,999 NT dollars	−0.048	0.590	0.058	0.516
50,000 NT dollars	−0.006	0.943	0.025	0.779
Employment				
Unemployed	Reference		Reference	
Unemployed due to disease	−0.024	0.790	0.112	0.211
Employed change due to disease	−0.023	0.799	−0.033	0.712
Employed	0.068	0.448	−0.005	0.955
Disease-related characteristics				
Duration of cancer diagnosis (year)	0.109	0.223	0.153	0.088
Cancer stage				
I	Reference		Reference	
II	0.018	0.840	0.091	0.309
III	−0.144	0.108	−0.232	0.010
IV	0.099	0.269	0.046	0.607
Comorbidity				
No	Reference		Reference	
Yes	0.028	0.757	0.000	1.000
Performance status				
ECOG = 0	Reference		Reference	
ECOG = 1	−0.017	0.847	−0.215	0.017
ECOG ≥ 2	0.009	0.922	−0.043	0.630
Characteristics of treatment				
Treatment following cancer diagnosis				
Cancer operation ± other treatments	0.012	0.890	−0.038	0.670
Other treatments	Reference		Reference	
Symptom distress	−0.131	0.143	−0.546	<0.001

## Data Availability

The data presented in this study are available on request from the corresponding author. The data are not publicly available due to privacy.

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
