# Peer review of "Self-Esteem as a Predictor of Mental Adjustment in Patients with Breast Cancer"

_ijerph, 2021, doi:10.3390/ijerph182312588_

Round 1

Reviewer 1 Report

I would like to start by congratulating the authors and congratulating them for having decided to investigate an area where there is still so much to discover, but also for having decided to share this findings with the rest of the scientific community, so that science can evolve.

This is an article about the Self-Esteem as a Predictor of Mental Adjustment in Patients with Breast Cancer.

All comments, questions and suggestions presented are constructive and try to improve the article, after several careful readings.

Abstract

Objective presented in the abstract must be the same as defined at the end of the Introduction

I would like to see a more focused conclusion on clinical practice.

Keywords

Repetitions with expressions that are in the title should be avoided. Whenever possible, keywords should be Mesh.

Introduction

To what extent is this an innovative study? What can it bring new to clinical knowledge and decision-making?

The objective at the end of the Introduction must be exactly the same as that of the Abstract or vice versa.

The bibliographical review leaves a little to be desired, especially in a topic that has had so much evolution over the years. References from 1979, 1987, 1994, 1995 (2), 1999, 2002, 2004 (2), 2008, 2009, 2013, 2014, 2015 (2), 2017, 2019 (3). 2020 cannot be considered recent or justifications of the current state of the art and are not enough to reach the standards of this magazine. By the way, with these references is the feeling that this article will have already been submitted and rejected and not updated in this component.

Discussion and Conclusions

The same note about the bibliographical references, noted above.

General comments

Interesting article, with a clear potential for publication, with a very interesting approach and with the potential to change clinical decisions, although not reflected in the conclusion.

The bibliography presented is not in line with the quality of this journal.

The article as presented must submitted to major revision.

Author Response

Reviewer1#

I would like to start by congratulating the authors and congratulating them for having decided to investigate an area where there is still so much to discover, but also for having decided to share this findings with the rest of the scientific community, so that science can evolve.

This is an article about the Self-Esteem as a Predictor of Mental Adjustment in Patients with Breast Cancer. All comments, questions and suggestions presented are constructive and try to improve the article, after several careful readings.

Authors’ response:

Thank you very much.

Abstract

Objective presented in the abstract must be the same as defined at the end of the Introduction. I would like to see a more focused conclusion on clinical practice.

Authors’ response:

Thank you very much for the valuable comments. We have revised the Abstract and Introduction section. We also had a more focused conclusion on clinical practice. The revised parts were highlighted in blue and used track changes.

Keywords

Repetitions with expressions that are in the title should be avoided. Whenever possible, keywords should be Mesh.

Authors’ response:

Thank you very much for the valuable comments. We have added the Mesh term.

Introduction

To what extent is this an innovative study? What can it bring new to clinical knowledge and decision-making?

Authors’ response:

Thank you very much for the valuable comments. We have revised the Introduction section and highlighted in blue as well as used track changes

The objective at the end of the Introduction must be exactly the same as that of the Abstract or vice versa.

Authors’ response:

Thank you very much. We have revised the objective at the end of the Introduction as the same as that of the Abstract. We have highlighted in blue and used track changes.

The bibliographical review leaves a little to be desired, especially in a topic that has had so much evolution over the years. References from 1979, 1987, 1994, 1995 (2), 1999, 2002, 2004 (2), 2008, 2009, 2013, 2014, 2015 (2), 2017, 2019 (3). 2020 cannot be considered recent or justifications of the current state of the art and are not enough to reach the standards of this magazine. By the way, with these references is the feeling that this article will have already been submitted and rejected and not updated in this component.

Authors’ response:

Thank you very much for the comments. We have rechecked and updated the References to reach the standards of this magazine.

Discussion and Conclusions

The same note about the bibliographical references, noted above.

Authors’ response:

Thank you very much for the suggestion. We have revised in Discussion and Conclusion sections. The revised parts were highlighted in blue and used track changes.

General comments

Interesting article, with a clear potential for publication, with a very interesting approach and with the potential to change clinical decisions, although not reflected in the conclusion.

Authors’ response:

Thank you very much. We have revised the Conclusion section and highlighted in blue.

The bibliography presented is not in line with the quality of this journal. The article as presented must submitted to major revision.

Authors’ response:

Thank you very much for the valuable comments. We have revised the bibliography to improve the quality of this paper.

Reviewer 2 Report

The article entitled „Self-esteem as a predictor of mental adjustment in patients with breast cancer” is an interesting research report consistent with the current of the global strategy for fighting cancer. The research project has been prepared and carried out appropriately. The Introduction section includes the factual reason and shows precisely an area of the empirical gap in which the research fills. The research project, selection of the research group, application of the instrument along with factual reason for its choice do not have any reservations. The description of the research results is clear and synthetic.

The poorest part of the work is the Discussion section since it does not introduce much. It includes elements that are confirmed by the results but they are not innovative enough. It reflects and confirms a rational analysis of clinical practice experience. Generally, it lacks any proposal how to use the knowledge in the clinical practice and what changes should be introduce to improve the quality of life in breast cancer patients with the inclusion of a legal, organisational and interdisciplinary approach to care of this group of women.

The abstract says tentatively about „facilitating the early introduction of appropriate nursing interventions, consequently, leading to the improvement of both self-evaluation and mental adjustment in those patients” but this research area is typically psychological. Thus, the question arises if a nurse has appropriate competencies to boost patients’ self-esteem and build their mental adjustment. The solution of rights and nursing practice in Taiwan, where the research was performed, may have such justification but then such information should be given in the Background section.

The other weakness of the article is the selection of references; it is worth using more recent and current sources of knowledge. There are too many references from the 20th century:

*Courneya, K. S.; Friedenreich, C. M., Physical exercise and quality of life following cancer diagnosis: a literature review. Annals of Behavioral Medicine. 1999, 21, (2), 171.

*Branden, N., The six pillars of self-esteem. Bantam Doubleday Dell Publishing Group Incorporated: 1995. *Greer, S.; Watson, M., Mental adjustment to cancer: its measurement and prognostic importance. Cancer Surv. 1987, 6, (3), 439- 53

*Watson, M.; Law, M. G.; Santos, M. d.; Greer, S.; Baruch, J.; Bliss, J., The Mini-MAC: Further development of the Mental Adjustment to Cancer Scale. Journal of Psychosocial Oncology. 1994, 12, (3), 33-46

 * Lewis, M. S.; Gottesman, D.; Gutstein, S., The course and duration of crisis. Journal of Consulting and Clinical Psychology. 1979, 47, (1), 128

*Oken, M. M.; Creech, R. H.; Tormey, D. C.; Horton, J.; Davis, T. E.; McFadden, E. T.; Carbone, P. P., Toxicity and response criteria of the Eastern Cooperative Oncology Group. Am J Clin Oncol. 1982, 5, (6), 649-55

*Akechi, T.; Okamura, H.; Yamawaki, S.; Uchitomi, Y., Predictors of patients' mental adjustment to cancer: patient characteristics and social support. British Journal of Cancer. 1998, 77, (12), 2381-2385

*Schnoll, R. A.; Harlow, L. L.; Stolbach, L. L.; Brandt, U., A structural model of the relationships among stage of disease, age, coping, and psychological adjustment in women with breast cancer. Psycho-Oncology: Journal of the Psychological, Social and Behavioral Dimensions of Cancer. 1998, 7, (2), 69-77

Author Response

Reviewer2#

The article entitled “Self-esteem as a predictor of mental adjustment in patients with breast cancer” is an interesting research report consistent with the current of the global strategy for fighting cancer. The research project has been prepared and carried out appropriately. The Introduction section includes the factual reason and shows precisely an area of the empirical gap in which the research fills. The research project, selection of the research group, application of the instrument along with factual reason for its choice do not have any reservations. The description of the research results is clear and synthetic.

Authors’ response:

Thank you very much.

The poorest part of the work is the Discussion section since it does not introduce much. It includes elements that are confirmed by the results but they are not innovative enough. It reflects and confirms a rational analysis of clinical practice experience. Generally, it lacks any proposal how to use the knowledge in the clinical practice and what changes should be introduce to improve the quality of life in breast cancer patients with the inclusion of a legal, organisational and interdisciplinary approach to care of this group of women.

Authors’ response:

Thank you very much for the valuable comments. We have revised the Discussion section and highlighted in blue using Track changes.

The abstract says tentatively about facilitating the early introduction of appropriate nursing interventions, consequently, leading to the improvement of both self-evaluation and mental adjustment in those patients” but this research area is typically psychological. Thus, the question arises if a nurse has appropriate competencies to boost patients’ self-esteem and build their mental adjustment. The solution of rights and nursing practice in Taiwan, where the research was performed, may have such justification but then such information should be given in the Background section.

Authors’ response:

Thank you very much for the valuable suggestions. We have revised in the Background section.

The other weakness of the article is the selection of references; it is worth using more recent and current sources of knowledge. There are too many references from the 20th century:

*Courneya, K. S.; Friedenreich, C. M., Physical exercise and quality of life following cancer diagnosis: a literature review. Annals of Behavioral Medicine. 1999, 21, (2), 171.

*Branden, N., The six pillars of self-esteem. Bantam Doubleday Dell Publishing Group Incorporated: 1995.

*Greer, S.; Watson, M., Mental adjustment to cancer: its measurement and prognostic importance. Cancer Surv. 1987, 6, (3), 439- 53

*Watson, M.; Law, M. G.; Santos, M. d.; Greer, S.; Baruch, J.; Bliss, J., The Mini-MAC: Further development of the Mental Adjustment to Cancer Scale. Journal of Psychosocial Oncology. 1994, 12, (3), 33-46

 * Lewis, M. S.; Gottesman, D.; Gutstein, S., The course and duration of crisis. Journal of Consulting and Clinical Psychology. 1979, 47, (1), 128

*Oken, M. M.; Creech, R. H.; Tormey, D. C.; Horton, J.; Davis, T. E.; McFadden, E. T.; Carbone, P. P., Toxicity and response criteria of the Eastern Cooperative Oncology Group. Am J Clin Oncol. 1982, 5, (6), 649-55

*Akechi, T.; Okamura, H.; Yamawaki, S.; Uchitomi, Y., Predictors of patients' mental adjustment to cancer: patient characteristics and social support. British Journal of Cancer. 1998, 77, (12), 2381-2385

*Schnoll, R. A.; Harlow, L. L.; Stolbach, L. L.; Brandt, U., A structural model of the relationships among stage of disease, age, coping, and psychological adjustment in women with breast cancer. Psycho-Oncology: Journal of the Psychological, Social and Behavioral Dimensions of Cancer. 1998, 7, (2), 69-77

Authors’ response:

Thank you very much. We have revised the references.

Round 2

Reviewer 1 Report

The authors carried out an important review of their article. My opinion is that it should be accepted for publication.